# Behavior Problems and Social Competence in Fragile X Syndrome: A Systematic Review

**DOI:** 10.3390/genes13020280

**Published:** 2022-01-30

**Authors:** Olga Cregenzán-Royo, Carme Brun-Gasca, Albert Fornieles-Deu

**Affiliations:** 1Department of Clinical and Health Psychology, Universidad Autónoma de Barcelona, Carrer de Can’-Altayó, S/N, 08193 Bellaterra, Spain; carme.brun@uab.cat; 2Consorcio Corporación Sanitaria Parc Taulí Sabadell, Parc Taulí, S/N, 08208 Sabadell, Spain; 3Department of Psychobiology and Methodology of Health Science, Serra Húnter Fellow, Universidad Autónoma de Barcelona, Carrer de Can’-Altayó, S/N, 08193 Bellaterra, Spain; albertfornieles@gmail.com

**Keywords:** behavior problems, social competence, fragile X syndrome, autism, anxiety, aggressive, socialization, attention, withdrawn

## Abstract

Fragile X syndrome (FXS) causes intellectual disability and is the known leading cause of autism. Common problems in FXS include behavior and social problems. Along with syndromic characteristics and autism comorbidity, environmental factors might influence these difficulties. This systematic review focuses on the last 20 years of studies concerning behavior and social problems in FXS, considering environmental and personal variables that might influence both problems. Three databases were reviewed, leading to fifty-one studies meeting the inclusion criteria. Attention deficit hyperactivity disorder (ADHD) problems remain the greatest behavior problems, with behavioral problems and social competence being stable during the 20 years. Some developmental trajectories might have changed due to higher methodological control, such as aggressive behavior and attention problems. The socialization trajectory from childhood to adolescence remains unclear. Comorbidity with autism in individuals with FXS increased behavior problems and worsened social competence profiles. At the same time, comparisons between individuals with comorbid FXS and autism and individuals with autism might help define the comorbid phenotype. Environmental factors and parental characteristics influenced behavior problems and social competence. Higher methodological control is needed in studies including autism symptomatology and parental characteristics. More studies comparing autism in FXS with idiopathic autism are needed to discern differences between conditions.

## 1. Introduction

As one of the most frequent inherited reasons for intellectual disability (ID) [1,2], fragile X syndrome (FXS) is caused by silencing of the fragile X gene FMR1 due to large expansions of non-coding CGG repeats [3]. The trinucleotide expansion inactivates the FMR1 gene, resulting in an absence of the fragile X mental retardation protein (FMRP), which is fundamental for natural neural development [4]. Repeat sizes conditioning the development of the different phenotypes vary in the unaffected population from 6 to 50 repeats, while individuals with the premutation (PM) have repeat sizes between 55 and 200. As a result of genetic inheritance, these PM alleles tend to expand to a full mutation in family members [5]. Consequently, the members with the full mutation (FM) expansions, which implies more than 200 CGG repeats of the gene, have a silenced gene, resulting in the absence of the FMRP protein and the full development of FXS [2]. 

Until recently, individuals with PM alleles were believed to be psychologically unaffected [6]. However, several implications of this condition have been found, including a significant contribution to the risk of attention deficit hyperactivity disorder (ADHD), subtle white matter structural changes, diminished brain activation in the amygdala and several brain areas that mediate social cognition, and long-term verbal memory recall deficits [7,8,9,10]. In fact, via increased FMR1 mRNA production and toxicity, the PM alleles can produce a family of neurodevelopmental phenotypes (ADHD, autism spectrum disorder, seizure disorder) and neurodegenerative phenotypes (fragile X-associated tremor/ataxia syndrome) [3]. 

The FXS FM phenotype occurs in both genders, although males tend to show greater cognitive impairment than females [11,12], with considerable variability in the degree of ID [13]. Females with FXS are usually less affected due to the extra X chromosome that partially compensates for the problem of the affected chromosome [14,15]. However, variability in females is even greater, with 1/3–1/2 of FM females exhibiting normal intellectual functioning [16]. Recent estimates of the prevalence of males and females with FXS are around 1 in 3600 to 4000 and 1 in 4000 to 6000, respectively [17]. 

### 1.1. Comorbidities

FXS individuals show comorbidities with other disorders such as ADHD, autism spectrum disorder (ASD), and anxiety [18]. ADHD estimates of comorbidity have been established at 73% [19], with almost 60% of juvenile cases continuing in adulthood [20]. ADHD symptoms are the most prevalent recognized behavior problem in FXS for the majority of boys and many females [19,21,22,23], hindering social relationships at both home and school [24]. Regarding ASD, mutations in the FMR1 gene are a contributing cause to ASD as a part of the broader FXS phenotype [2]. Recent studies have stated a prevalence of comorbid FXS+ASD that varies from 30% to 67% [25,26,27,28]. Moreover, it has been estimated that among the population with autism (Aut), 2–6% of children have FXS [29]. Around 90% of individuals with FXS present atypical behaviors, as do nonsyndromic ASD individuals, such as motor stereotypies (i.e., hand flapping), self-injury, eye avoidance, or social avoidance [30,31,32,33]. However, controversy remains about the overlap between ASD and FXS [34]. Some authors claim that autism seen in FXS and idiopathic autism (IA) have considerable similarities, sharing changes in the neurobiology of facial emotion processing, with individuals with IA and comorbid individuals with FXS and autism showing behavioral problems similar to those of individuals with IA [18]. In contrast, others state that there are substantial differences, making comorbid subjects more vulnerable, with greater communication and social reciprocity impairments and higher levels of repetitive and challenging behaviors than individuals with FXS only [35], and with different neurobiological substrates of the behavioral impairments [36]. Comorbid individuals with FXS and autism have shown outcomes inferior to those of individuals with FXS without autism [26,37]. Last, regarding anxiety, symptoms are recognized as an outstanding feature of the phenotype of individuals with FXS [38], with 70% of males and 56% of females receiving treatment for anxiety symptoms or with a comorbid anxiety diagnosis [39].

### 1.2. Behavior Problems

Behavioral implications in FXS have been explored since 1943, with the description of Martin and Bell [40]. A particular behavioral phenotype, understood as a greater probability of exhibiting particular behavior due to the syndrome, has been observed for individuals with FXS, although there is some variability in behavioral symptoms [41,42]. They include cognitive difficulties, language problems, social anxiety, gaze aversion, hand stereotypies [31], repetitive and self-injurious behavior [43], and aggressive behavior [44], in addition to autistic-like features such as motor stereotypies and perseverative behavior [45]. A standard indicator of the intersection between ASD and FXS is repetitive behaviors [46]. In this regard, the same behaviors are shared in FXS and ASD phenotypes, although there is evidence signaling different phenotypes between FXS individuals and idiopathic autism (IA) [47]. Other authors have also found similar behavior difficulties in both groups [48]. Thus, determining behavioral phenotypes is valuable for identifying individuals with FXS at the early stages and for starting interventions and assessments as soon as possible [49]. However, there is still a need to address whether there are different phenotypes of FXS concerning behavior problems or self-injury, and how they develop as individuals get older, and specifically in transition stages at school and when entering employment [50]. Furthermore, scarce studies have focused on trajectories of behavior problems across adolescence and adulthood in individuals with FXS [51].

### 1.3. Social Skills and Social Competence

Social competence is a broad construct referring to adequately dealing with the demands of a social situation [52]. Four components could be addressed when considering this construct: peer relations, social cognition, behavior problems, and effective social skills [53]. Social skills could be defined as abilities associated with the development that contributes to the general level of social competence, including perspective taking, interpersonal problem solving, moral judgment, self-control, and communication facility [54]. To this effect, social skills are particular behaviors exhibited by an individual to be competent in a social task [55]. Since the ID definition includes both limitations in intellectual functioning and adaptive behavior in daily social and practical skills [56], it is not striking that individuals with FXS show deficits in social competence. As described above, autistic features, social anxiety, and pragmatic deficits in language are part of the social deficits included in the full mutation phenotype [57]. People with FXS are particularly characterized by social avoidance [23], with one of their most prominent features being eye contact avoidance, finding this an aversive stimulus, which is associated with changes in skin conductance, cortisol reactivity, and pupillary reactivity [58,59,60]. Social withdrawal is also considered part of FXS individuals’ clinical profile [61]. Considering the fact that ASD individuals show poor social skills as a prominent feature, with impairments in social communication, it is understandable that comorbid individuals with FXS and ASD are more avoidant than individuals with FXS without autism comorbidity [43,62]. In the same line, anxiety and autism symptoms have been found to be risk factors for reduced social skills in individuals with FXS [34]. Similarly, autism symptoms have been associated with reduced socialization skills in FXS [63].

### 1.4. Environmental Factors

ID was considered a lifelong disability of an individual’s characteristics until 1992 when the American Association on Intellectual and Developmental Disabilities stated a definition that considered environmental supports to improve individual functioning [56]. Considering ID as a state of functioning allowed discrepancies between person and environment to be considered when conducting ID studies [64]. In this scenario, providing support to individuals with ID could enhance their functioning in their environment, leading to a fuller life [56]. Supports are understood as assets and strategies that strive to facilitate the developmental and learning processes and interests and quality of life of persons trying to improve personal functioning. [65]. Moreover, changes in the family, educational, or home environment might help reduce or increase behavior problems in FXS [66]. However, parenting a child with FXS can be very challenging due to their behavioral phenotype, and even when they try to do their best, parents can be defeated trying to maintain a responsive parenting style [67].

Additionally, challenging behaviors in individuals with FXS are known to impact family functioning such as the mother’s mental health [68]. Some studies point to a higher susceptibility to stress for mothers with the PM condition [69,70]. Consequently, a highly stressed family environment could negatively influence the development of the offspring’s self-regulation and social competence, subsequently affecting the entire family system bidirectionally [71], with the family environment potentially affecting a child’s social and emotional functioning, and the child’s behavior possibly influencing their parents. Thus, bidirectional effects are well recognized in the general literature [72]. 

### 1.5. Current Study

This is a systematic review of the last 20 years of research focusing on behavior problems and social competence problems in individuals with FXS. Specifically, we want to address phenotypic differences between individuals with FXS, individuals with comorbid FXS and ASD, and IA individuals. Although a behavioral phenotype has been identified, there is a need to clarify the differences between individuals with FXS and ASD and nonsyndromic ASD individuals in terms of anxiety, manic/hyperactive behavior, and obsessive compulsive behavior [73]. Further, it seems that important information concerning differences between the FXS spectrum and IA might be masked by just relying on a categorical diagnosis of ASD [36]. Furthermore, other comorbidities contributing to both behavior problems and social competence are considered. Environmental factors that contribute to the observed problems in social competence and behavior problems are also addressed in this review since they might influence both variables. The specific questions addressed by this review are as follows:What behavior problems have been researched in individuals with fragile X syndrome in the last 20 years?What social competence problems have been researched in individuals with fragile X syndrome in the last 20 years?What differences have been found in behavior problems and social competence when comparing individuals with fragile X syndrome with typically developing individuals (TD) and individuals with other IDs?What differences have been found in behavior problems and social competence when comparing individuals with fragile X syndrome with comorbid individuals with fragile X syndrome and autism?How might environmental factors affect behavior and social problems in individuals with fragile X syndrome?

## 2. Materials and Methods

This review follows the Preferred Reporting Items for Systematic Reviews and Meta-Analyses (PRISMA) [74]. It has been registered in the PROSPERO database with ID number 284267, although due to the current pandemic, it has not yet been assessed by the resident professionals.

### Search Strategy

Three databases were searched: PsycINFO, PubMed, and Web of Science. The exact string search included “Fragile X Syndrome or FXS” in the title or in the abstract and the title depending on the defaults of the database, and “social skills OR social abilities OR social interaction OR social behavior OR social behaviour OR interpersonal skills OR social functioning OR social competence OR socialization OR problem behavior OR problem behaviour OR disruptive behavior OR disruptive behaviour OR dysfunctional behaviour OR dysfunctional behavior OR challenging behavior OR challenging behaviour OR behavioral problems OR behavioural problems OR externalizing OR internalizing OR aberrant behavior OR aberrant behaviour OR phenotype OR phenotypes OR maladaptive behavior OR maladaptive behaviour.” A preliminary search was conducted on 1 December 2019, and the main search was carried out on 1 October 2021. The databases were last searched on or before 10 October 2021. Term selection was carried out using MesH terms of PubMed and by consulting studies in the bibliographies related to social competence and behavior problems. Filters included in the searches were the date, only including studies from 2000 onwards, journal articles, and Spanish and English languages.

The inclusion criteria for papers to be considered in the review, apart from the filters included in the search, were as follows. The study had to focus empirically on behavior problems or social competence in individuals with FXS. Nevertheless, studies focusing on other variables were considered if they addressed behavior problems or social competence in their results. Only full-text papers were included, meaning that if a study could not be wholly retrieved, it would be excluded from the review. Moreover, the studies had to focus on individuals with FXS with the full mutation. The exclusion criteria were documents other than original research such as reviews, congress abstracts, books, single case studies, studies focusing on participants with IDs other than FXS (if the study addressed different groups, this exclusion criterion did not apply), and studies focusing on other phenotypes such as premutation, and comorbidity of individuals with FXS and fragile X-associated tremor/ataxia syndrome (FXTAS) or dementia. This search strategy led to a total of 1538 papers eligible for inclusion. After checking for duplicates, 785 papers were considered.

The titles and abstracts of the 785 papers that remained after checking for duplicates were screened by a researcher. An Excel matrix was developed to record why each study was excluded according to the exclusion criteria. Different categories were developed, and labels explaining the reason for exclusion were assigned. These were case studies, language, reviews, interventions, studies focusing on drug treatments, animal models, brain/metabolism (including neurotransmitters, hormones, cognitive functions, or neuroimaging studies), validation tools, other phenotypes (the premutation condition and FXTAS), and a category to include the articles that could not be categorized in the others called “Not in line with the topic” (sleep problems, epilepsy, supplements, dental studies). All the articles were thereby classified into categories according to their title and abstract, leaving 143 papers. A researcher assessed the full text of the 143 papers, leading to 64 more being excluded, leaving 79 papers (51 included in the review and 28 with lower scores).

The 79 papers were reviewed using a matrix adapted from another study, which assesses the quality of the studies that addressed phenotypes [49]. The only modification made to the matrix was in the second line, corresponding to autism comorbidity or symptom control in individuals with FXS. The researcher assessed all the documents twice within two weeks. If discrepancies were found between the assessments, two other researchers decided what score was most fitting. The matrix and discrepant decisions are depicted in Appendix B, Table A1. The discrepant decisions are indicated in Table A1 with an asterisk (*) when a study was correspondingly discrepant. After assessing the 79 papers using the matrix, only those in the upper third were included in the review, leaving the studies that scored 12 or more points. However, since most of these studies pertained to the social competence category (*n* = 26), and only 14 studies pertained to behavior problems, studies with scores of 11 and 10, which belonged to the behavior problems category, were also included in the review. This strategy led to 26 studies assessing social competence and 25 studies assessing behavior problems. The data from the studies included in this review can be found in Appendix A. The PRISMA flowchart depicted in Figure 1 is a visual description of the process of filtering the initial 1538 papers down to the 51 reviewed.

## 3. Results

### 3.1. Researched Profile in Behavior Problems

The main behavior problems researched represented a consistent behavioral phenotype in FXS, including attention deficit, hyperactivity, impulsivity, anxiety, repetitive, perseverative, stereotypic behaviors, affect, aggression, and self-injurious behavior [75]. The prevalence of the behavior problems indicated in the studies reviewed is summarized in Appendix B, Table A2. As expected in light of previous reviews, the highest prevalence and percentage scores of clinical concern were found for attention problems and ADHD comorbidity in boys and girls aged between 4 and 30 years [76]. The percentages of this behavior in the clinical range across studies varied from 15 to 73.5% in boys, as the table shows. The lowest percentages in the clinical range corresponded to the samples with the highest age range [77,78]. Furthermore, almost all the boys with FXS aged between 5.7 and 16.1 years were inattentive and easily distracted (98–100%, respectively), followed by over-active and impulsive [79]. Supporting these findings using a mood scale, the highest scores for boys between 4 and 10 years were for manic/hyperactive [73]. 

Less prevalent behavior problems of clinical concern for individuals aged between 4 and 12 years were thought problems and withdrawn and aggressive behavior problems in girls, withdrawn and aggressive behavior in boys [66], and aggressive behavior in boys and girls at 6 years [78]. Depression problems were the least reported psychiatric concern, and their prevalence was very low in children and low in adolescents (1.2–16%, respectively) [63], although these percentages were slightly higher in other studies [73,78]. 

Four of the studies reviewed contribute with results on disruptive behaviors. In boys aged between 11 and 18 years, stereotypy showed the highest prevalence and happened most often with a daily median frequency, although it was the least severe behavior problem [80]. Aggression was the most severe reported behavior problem by parents, followed by property destruction, which was also reported in the moderate zone but as the least prevalent behavior. Both behavior problems exhibited a weekly median frequency. The daily prevalence for aggression was between 20% and 40%. A slightly higher daily prevalence (50%) for aggression was found for boys in the same age range, and 25% of caregivers reported this behavior as a significant threat to health and safety [77]. In an older sample of males aged between 6 and 47 years, 21.5% exhibited persistent aggressive behavior, and higher impulsivity scores were associated with an increased probability of exhibiting aggressiveness over time [81]. Aggression was also found to be the most prevalent behavior problem and was the behavior problem with fewer boys not showing it between 5 and 21 years old [82].

Regarding self-injurious behavior, over eight years, almost 50% of boys aged between 16 and 25 years exhibited this behavior, with 32.4% of them exhibiting it persistently [81]. Higher scores in restricted, repetitive, and stereotyped behaviors predicted an increased probability of exhibiting continuous self-injury behavior across all the assessment points. A higher percentage (70.6%) of self-injurious behavior was found in individuals aged between 11 and 18 years [80]. However, in boys aged between 6 and 10 years, self-injury was the least reported behavior problem and was also less problematic [42].

#### Developmental Trajectories in Behavior Problems

Some studies have found decreasing trajectories of behavior problems such as physical aggression and tantrum scores until age 19 years, although they were not associated with age after this point [83]. The prevalence and severity of aggression also declined from 40% to 20% and 30% to 10% in boys aged between 11 and 12 years and 17 and 18 years [80], respectively, and a decreased proportion of aggressive behaviors in the clinical range was found from age 6 to 18 years in boys and girls [78]. Moreover, self-injury frequency was lower in boys aged between 17 and 18 years than in the age group 11–12 years [80], and boys and girls showed decreased percentages of attention problems in the clinical range from age 6 to 18 years [78]. Furthermore, higher rates of total behavior problems were found in both children and adolescents than in adults [84]. However, stability in aggressive behaviors in males aged between 6 and 47 years over 8 years has also been found [81].

Other behavior problems showed stability in longitudinal studies, including verbal aggression in males and females from 5 to 40 years [83], total problems assessed with the Child Behavior Checklist [85] in boys aged between 4 and 12 years [22], externalizing behaviors over three years in males and females aged between 12 and 48 years [86], and total and externalizing behaviors in a cross-sectional sample when comparing boys under 10 years old and boys over 11 years old [87]. No difference by age was found in attention/hyperactivity or opposition in individuals with FXS aged between 3 and 30 years [77]. Over-activity and impulsive speech remained stable in individuals aged between 6 and 54 years [88]. Trajectories of behavior problems are visually described in Appendix B, Figure A1.

Trajectories of behavioral problems were affected by ASD comorbidity in two longitudinal studies. Decreases in attention problems and aggressive behaviors in individuals aged between 6 and 18 years were smaller for those with autism comorbidity than those with FXS only [78]. Moreover, in individuals aged between 6 and 54 years, impulsivity and repetitive questioning scores decreased only for the group with low levels of autistic symptomatology, and not for the comorbid FXS+ASD group [88].

### 3.2. Researched Profile in Social Competence

The researched social competence profile is less unitary than that for behavior problems. Studies have mainly assessed social behavior profiles, social avoidance, social approach behavior, and many specific variables that do not respond to a unitary theoretical approach to social competence, but rather to specific behaviors associated with it. The studies switched between parent-reported measures and task measures to address this area. Concerning prevalence, 81% of males exhibited social avoidance between ages 4 months and 25 years [89]. Regarding problems associated with social competence in the clinical range, between 15 and 35% of individuals with FXS aged between 3 and 30 years experienced social issues, anxiety, and adaptive social problems in the clinical range [77]. Withdrawal problems appeared in 21.5% and 17.5% of the sample of boys and girls, respectively, between the ages of 6 and 17 years in [66], and for 17% of boys aged between 4 and 12 years in [22]. Social problems appeared in the clinical range for 40% and 41.8%, respectively, of boys and girls [66] and for 26% of boys [22]. 

#### Developmental Trajectories in Social Competence

Some studies found positive trajectories for social competence over the years in individuals with FXS. These included reduced discomfort in older individuals with FXS when conducting a social task in a sample aged between 6 and 17 years [59], positive correlations between chronological age (CA) and social responsiveness in boys with a mean age of 15 years [90], higher scores in social skills in older individuals in a sample of boys aged between 3 and 7 years [34], and a positive correlation between social motivation and age in boys with FXS with a mean age of 23 years [91].

Regarding socialization, the longitudinal studies reviewed found different trajectories across ages. Significant declines between ages 2 and 14 years, which stabilized between 14 and 18 years, in boys were found, in addition to stability for girls [91]. However, in studies controlled by ASD symptomatology, boys aged between 1 and 12 years steadily improved their socialization scores [92]. Increasing socialization was found between ages 2 and 9.5 years, which were restrained around 7.5 years, with many infants tending to show declines at this age [93]. Furthermore, socialization scores decreased over three years in boys with a mean age of 4.7 years [94]. 

Internalizing symptoms were stable over three years in males and females aged between 12 and 48 years, although older individuals showed slightly fewer internalizing symptoms [86], and they were likewise stable in a cross-sectional sample of boys aged under 10 years and boys over 11 years old [87]. However, in another cross-sectional study, a higher frequency of internalizing symptoms in adolescents than in children was found [84].

Regarding social anxiety and depression, studies have found stability or worsening status. Anxious/depressed scores in girls with FXS aged between 10 and 15 years were found to be stable over three years [95], as were scores in boys and girls aged between 6 and 18 years [78]. Depression and anxiety disorders were also found to be stable until 30 years old [77]. However, social anxiety correlated negatively with age in boys with a mean age of 23 years, and a positive association for social anxiety with age was found in boys, with a mean age of 15 years [90,96]. 

Other studies have found stability in different variables, including social issues in individuals aged between 3 and 30 years [77] and withdrawal in girls aged between 10 and 15 years over three years [95]. However, in younger boys with a mean age of 4.7 years, increases in withdrawn behaviors with age were found over four years of assessments [93]. Higher increasing avoidance in initial interactions with unfamiliar people has been seen in early childhood (4–72 months) in terms of eye contact, physical approach, and facial expressions, but not in familiar interactions. However, in adolescent and adult samples (10–25 years), social avoidance was not associated with age [89]. Supporting this finding, eye gaze avoidance in boys between 13 and 22 years was stable [97]. Trajectories of social competence are visually described in Appendix B, Figure A2.

Three studies found autism and anxiety symptomatology to be interaction variables affecting trajectories in social competence. Boys aged between 3 and 14 years with low levels of autism symptomatology showed a significant increase in social skills with age. Those with middle levels of ASD reached higher scores in total social skills earlier in life, showing reduced increases over the years, and those with high levels showed meager total social skills at younger ages, increasing minimally over time [34]. Moreover, girls with FXS with higher scores in autistic symptomatology obtained poorer outcomes and slower development rates in the personal social domain than those with lower scores [98]. Furthermore, higher ASD symptomatology was associated with a slower growth rate in socialization scores in boys at 24 months, while there was no association up to this age [99]. Regarding anxiety, boys with low and medium anxiety symptoms showed significant increases in social skills with age, while the group with higher anxiety scores showed minimal increases in social skills over the years [34]. 

### 3.3. Differences in Behavior Problems and Social Competence, Comparing Individuals with Fragile X Syndrome with Individuals with TD and Individuals with Other IDs

#### 3.3.1. Comparison of Behavior Problems between Individuals with FXS and Individuals with TD

The studies found greater behavior problems for individuals with FXS than for TD individuals, including more significant total, internalizing, and externalizing behaviors in boys between 5.7 and 16.10 years with FXS [87]. Boys and girls with FXS with a mean age of 10.9 years showed higher scores in externalizing behaviors than their unaffected siblings [100]. Significantly higher levels of irritability, stereotypic behavior, inappropriate speech, hyperactivity, more symptoms on the attention and hyperactivity scale, and more issues in the repetitive behavior questionnaire were found, in addition to a trend for higher opposition scores in individuals with FXS aged between 3 and 30 years [77]. Furthermore, compared to normative data, adolescents and adults with FXS scored above the clinical cut-off in behavior problems [35]. Interestingly, boys aged between 5 and 8 years with FXS exhibited significantly higher hyperactivity and lower attention scores than CA-matched and mental age (MA)-matched TD boys with good attentional abilities, but they did not differ from CA-matched and MA-matched TD boys with deficient attentional abilities [101]. 

In some behavior problems, no differences were found between individuals with FXS and those with TD. For boys aged between 6 and 17 years, there were no differences for somatic complaints and delinquent behaviors [66]. No difference was found in somatic complaints and delinquent behavior in boys aged between 4 and 12 years [22], or in aggressive behavior in girls and boys aged between 6 and 17 years [63]. Furthermore, no differences were found in delinquency and aggressive behaviors in boys with FXS and TD individuals with deficient attentional abilities aged between 8 and 15 years [101].

#### 3.3.2. Comparisons of Behavior Problems between Individuals with FXS and Individuals with Other IDs

Regarding Down syndrome (DS) groups, boys aged between 5 and 18 years with FXS scored significantly lower in attention abilities and had higher hyperactivity than the DS group [101]. Significantly higher scores on stereotyped behaviors, repetitive behaviors, insistence on sameness, impulsivity, and over-activity were found for the boys with FXS aged between 6 and 39 years compared to the DS group, but no differences were found in compulsive behavior [102].

Regarding individuals with other IDs, fewer differences were found. There were no differences in the severity or frequency of aggression for boys with FXS between the ages of 11 and 18 years compared to those with other intellectual and developmental disabilities (IDD) matched by CA [103]. There were no differences in the frequency of aggression in boys in the same age range compared to a mixed etiology group, although the aggressive behaviors of the FXS group were rated as less severe [80]. In this study, stereotypy was also more prevalent and self-injury more frequent in individuals with FXS, but both groups showed the same relative frequency in property destruction and stereotypy. In a study comparing boys with FXS aged between 3 and 6 years and boys with a developmental delay of unknown etiology, no differences were found in general activity, task orientation, attention problems, hyperactivity, rigidity (difficulties with changes), somatic complaints, and irritability [104]. Even after controlling for maternal persistence, distractibility, and irritability, there were no differences in attention, hyperactivity problems, and task orientation. However, when maternal ratings of their activity level were controlled for, a higher general activity level was found for the FXS group [104]. However, activity levels did not differ in males aged between 6 and 39 years compared to individuals with Phelan–McDermid syndrome [102]. This study found a difference in total repetitive, with these behaviors being higher for FXS individuals. Additionally, boys aged between 5 and 16 years showed similarities in the frequency of behavior problems to boys with fetal alcohol syndrome (FAS), particularly in disruptive behaviors such as irritability, abusive/swearing at others, and attention/seeking, as well as overexcited/impulsive, and flick, tap, and twist objects [79]. At the same time, other behavior abnormalities, including autistic-related behaviors such as eye avoidance and self-absorbed behaviors, were more frequent in boys with FXS than in individuals with FAS, tuberous sclerosis, or Prader–Willi syndrome [79].

Comparisons with ASD samples showed no differences or greater behavior problems for FXS groups. There were no differences in the activity level, repetitive behaviors, or compulsive behaviors in males aged between 3 and 39 years [102], or in internalizing and externalizing behaviors in males with a mean age of 21 years [35]. A trend for higher manic/hyperactive behavior was found for FXS boys aged between 4 and 10 years compared to CA-matched boys with ASD. This trend increased until reaching significance when controlled by intellectual quotient (IQ) and ASD symptomatology [73]. A trend for lower obsessive compulsive behaviors for FXS appeared but did not remain when controlled by ASD and IQ. Additionally, in the early stages, 36–95 months, boys with FXS were significantly more distractible than those with an autism diagnosis (AD) [105].

#### 3.3.3. Comparisons of Social Competence between Individuals with FXS and Individuals with TD

A worse profile in social competence was generally found for individuals with FXS than for TD individuals, including lower socialization scores than CA-matched TD boys aged between 3 and 13 years [106], lower adaptive socialization scores in boys over 12 years and in girls aged between 2 and 18 years [91], and a lower growth rate in social development in individuals with FXS compared to TD references aged between 12 and 143 months and in boys aged between 6 and 24 months [92,99]. Higher percentages of social avoidance (eye contact, physical approach, and facial expression) at initial assessments and in the last hour of assessment for boys aged between 4 and 72 months were also recorded [89]. Similar levels of physical approach in the last hour of assessment were found between FXS individuals with a mean age of 4 years and TD individuals [107]. A significantly higher mean proportion of eye avoidance was found in boys with an average age of 16.52 years [97], in addition to impaired eye contact, vocal quality, increased discomfort, and task avoidance during a social task in individuals aged between 6 and 17 years [59]. One study recorded lower scores in social skills in boys with FXS aged between 3 and 7 years [34], and another found more withdrawal and adaptive behavior in a socially desirable way that developed more slowly in boys with FXS aged between 36 and 95 months [105]. One study found significantly lower social scores in adaptive behavior, more significant difficulties with social issues, significantly higher levels of lethargy, and higher anxiety scores in individuals with FXS aged between 3 and 30 years [77]. Girls with FXS aged between 7 and 18 years exhibited significantly greater impairments in identifying the causes and consequences of social problems, and a trend for generating less competent goals and solutions to social problems and performing at lower rates in all social information processing tasks, compared to TD girls [108]. Higher anxiety levels, withdrawal, and social problems in individuals with FXS than in CA- and MA-matched TD individuals with both good and impaired attentional abilities have been found [101], as have lower facial fear expressions of avoidance compared to TD individuals, resulting in an untypical response to unknown people [109]. 

However, some behavior problems did not differ among individuals with FXS and TD individuals, including anxiety/depression scores in boys aged between 6 and 17 years [66], and escape behaviors and distress vocalization when exposed to unknown individuals across years in individuals with a mean age of 38 months [109].

#### 3.3.4. Comparisons of Social Competence between Individuals with FXS and Individuals with Other IDs

A more impaired profile is seen in individuals with FXS than in individuals with DS, including lower socialization skills in boys with FXS aged between 3 and 13 years [106], more eye gaze avoidance in a sample aged between 13 and 22 years [97], and higher social anxiety in boys with a mean age of 23 years and boys aged between 5 and 18 years [96,104]. However, in the social composite, a lower percentage of individuals with FXS between 3 and 30 years were in the clinical range [77]. However, differences in social motivation were not found in boys with a mean age of 23 years [96].

Comparisons with individuals with other IDs found similar anxiety/depression scores to individuals with a developmental delay of unknown etiology [104], and similar socialization scores for individuals aged between 5 and 21 years compared to Smith–Magenis syndrome (SMS) and non-specific intellectual and developmental disabilities (NSID) [86]. Furthermore, no differences were found in social motivation in boys with a mean age of 23 years compared to individuals with Cornelia de Lange syndrome (CdLS) and Rubenstein–Taybi syndrome (RTS) [96]. However, significantly more avoidance (to novel situations) and less withdrawal were found in boys aged between 3 and 6 years than in individuals with a developmental delay of unknown etiology [104]. Nonetheless, when controlled by maternal temperament (general activity level and sleep, approach/avoidance, flexibility/rigidity, mood, persistence, and distractibility), the boys with FXS were reported to only have a trend of increased avoidance/approach behavior compared to the boys belonging to the control group, but still exhibiting significantly less withdrawn behavior. Lower eye contact and focus of attention were found in boys with FXS compared to individuals with CdLS but similar levels of eye contact to individuals with RTS in [90]. 

Compared to individuals with autistic disorders (ADs), a better profile of social competence appears together with higher anxiety levels. Adolescents and adults with FXS have been shown to be almost 12 times more likely to have a mutual friend than those with AD, and when accounting for behavior problems, it seems that the diagnostic group has no influence on friendship, while the level of behavior problems does [110]. Moreover, individuals with FXS were significantly more likely to have a hobby and spend time with friends and neighbors than individuals with AD. One study showed that adults with FXS were more engaged in recreational activities and playing sports than adults with AD, but no differences were found between the adolescent samples. The two groups of teenagers and adults did not differ in time spent with coworkers, attending religious services, social events, or religious groups, or travel. Male infants aged between 36 and 95 months with FXS had a higher developed personal-social domain than AD males (7.8 months lower) [105]. However, higher mean scores in general anxiety were found for individuals with FXS aged between 4 and 10 years than CA-matched ASD individuals even when controlled for verbal IQ and ASD symptomatology [73].

### 3.4. Differences in Behavior Problems and Social Competence between Individuals with FXS Only and Those with Comorbid ASD

#### 3.4.1. Differences in Behavior Problems

Five studies comparing groups found greater behavior problems for individuals with FXS+ASD than for those with FXS only from childhood to young adulthood. These included higher attention problems [35,46,48,111], internalizing [35,46,48], hyperactivity/over-activity [88,111], aggressiveness or irritability/aggressive behaviors [35,111], repetitive or stereotyped behavior [46,48,88], total problems [35,46], externalizing, thought problems, intrusive thoughts, socially offensive behaviors, uncooperative behaviors, and being more hurtful to others in a sample with a mean age of 21 years [35]. Higher perseverative/obsessive compulsive behavior has been recorded in a sample aged between 0 and 21 years [111], as well as higher impulsivity, lining up, and just right behavior over 19 years old in a sample aged between 6 and 54 years [88]. However, no differences were found between samples in other behavior problems such as destructive to property, hurtful to self, disruptive behaviors, unusual habits [35], and irritability [48]. 

Furthermore, no differences in behavior by group (FXS or FXS+ASD) were found in a study with a sample group of over 200 individuals aged between 3 and 11 years and over 12 years, although this study assessed comorbidities by requesting information from parents and not by using a validated measure to establish the groups [63]. The only slight difference was higher self-injury behavior in the FXS+ASD group, but there were no differences in attention, hyperactivity, or aggressive behavior.

Three studies comparing behavior problems in individuals with FXS+ASD and those with ASD only showed different results that could be age related. A study conducted on young adults (mean age 21 years) with comorbid FXS+ASD showed higher scores in challenging behavior, total behavior problems, socially offensive behavior, uncooperative behavior, intrusive behaviors, and attention problems. The two groups did not differ significantly in aggressive behaviors, thought problems, externalizing problems, hurtful to self, property destruction, and disruptive behavior. However, the ASD group exhibited significantly lower strengths than the FXS+ASD group [35]. In a study with a juvenile sample (3–5 years), the groups did not differ in stereotypy, self-injury, and insistence on sameness, with the autism group only showing higher impairments in compulsive and ritual behaviors than the FXS+ASD sample [47]. Last, a study comparing young individuals (mean age 4.6 years) with FXS+Aut and those with developmental language delay (DLD) and Aut showed significantly higher scores in the borderline and clinical range in total behavior problems, thought, attention, and aggressive problems [46].

#### 3.4.2. Differences in Social Competence

A more impaired profile in social competence is seen in individuals with comorbid FXS+ASD than in those with FXS only in the studies reviewed, including significantly lower socialization scores in boys with FXS+ASD compared to boys and girls with FXS only aged between 12 and 143 months [92], and lower socialization scores in a small sample of boys aged between 21 and 48 months [26]. A lower developed personal domain in boys aged between 36 and 59 months was also seen [105]. Lower socialization and lethargy/social withdrawal over two years were observed in the FXS+ASD group in individuals with a mean age of 4.7 years, but these differences did not remain in older ages [94]. Higher social withdrawal, withdrawal/lethargy, and greater delays in socialization were seen for boys with FXS+ASD aged between 3 and 8 years [112], as well as higher withdrawal and lethargy/social withdrawal in boys with an average age of 4.7 years [46]. Higher withdrawn/inattentive behaviors were also seen in comorbid FXS+ASD with a mean age of 21 years than in individuals with FXS only [35]. Supporting these results, males and females aged between 2 and 9.5 years with FXS only showed the sharpest increase in socialization scores over the years compared to those with FXS+Aut [93]. Furthermore, in boys aged between 4 and 5 years, increasing impairments in social interaction have been found in the groups FXS only, FXS + pervasive developmental disorder (PDD), and FXS+ASD, in all measures corresponding to the social behavior profile (withdrawal, lethargy/social withdrawal, socialization, and daily living) [48]. Individuals aged between 5 and 16 years with comorbid FXS+ASD also made significantly fewer non-comprehension signals (more impairment) than the FXS-only group during a social task [113]. Higher anxiety scores for comorbid individuals aged between 12 and 21 years [111] and higher impairments for social information processing tasks (problem identification, goal generation, first solution competency, and chosen solutions) in comorbid girls with FXS+ASD than in FXS-only girls aged between 6 and 17 years were also found [108]. Furthermore, although FXS-only individuals acted similar to those with comorbid FXS+ASD in social approach during the first minute of interaction, the comorbid group exhibited less social approach (more avoidant) in the last hour of assessment [107]. However, in other issues such as mood swings/depression, there were no differences between individuals with FXS only and those with comorbid ASD [111].

Special attention must also be paid to studies comparing comorbid FXS+ASD and individuals with ASD only, although they do not provide a clear picture. In socialization, the FXS+ASD group achieved lower scores that disappeared when controlled for age and IQ [46], while in another study, the lowest personal-social levels were attributed to young boys with AD aged between 36 and 59 months compared to boys with FXS+ASD [105]. Furthermore, for withdrawal and internalizing, lower scores were found for the FXS+ASD group compared to the DLD+Aut group, unlike other studies that have found greater internalizing problems, while non-significant and greater withdrawal/inattentive problems for the FXS+ASD group have also been reported [35,46].

### 3.5. Environmental Factors Affecting Behavior and Social Problems in FXS

#### 3.5.1. Environmental Factors Affecting Behavior Problems

In several studies, three characteristics of the mothers were associated with behavior problems. Higher maternal criticism was associated with externalizing symptoms in children, adolescents, and adults and with total behavior problems in adolescents, and high criticism in families was associated with a higher severity of behavior problems in individuals aged between 12 and 48 years [84,86]. On the other hand, positive comments were associated with lower externalizing problems in children and adults and total problems in adolescents [84]. Higher maternal warmth was also associated with lower levels of externalizing problems in children and adults, total behavior problems in children and adults, and decreases in behavior problems in individuals aged between 12 and 48 years [84,86]. Another maternal characteristic, flexibility, was associated with attention problem scores, with individuals with more flexible mothers showing greater declines in attention problems over the years [78], while warmth and affect were not significant in this study. This characteristic of the mother combined with ASD comorbidity led to four possible outcomes in an individual’s attention problems. Similar scores were achieved by individuals with FXS+ASD with mothers with high flexibility and FXS-only individuals with mothers with low flexibility. Both exhibited medium decreases in attention problems over the years. In contrast, individuals with comorbid ASD and mothers with low flexibility maintained attention problems over the years, and individuals with FXS only with highly flexible mothers showed the highest decreases in attention problems over the years. Higher maternal educational level, apart from predicting total behavior problems, was associated with higher reports of behavior problems and attention problems in their offspring aged between 4 and 12 years [22]. 

Parental psychopathology was associated with internalizing and externalizing behaviors in boys and girls with FXS, respectively, between 6 and 17 years [66]. However, maternal distress was not associated with behavior problems in individuals with a mean age of 10.9 years [100]. 

Other positive aspects of the environment (cohesion, expressiveness such as sharing personal problems, achievement orientation, active recreational activities, independence, intellectual-cultural, moral-religious, control as rules, and organization) were not related to behavior problems in individuals with FXS with a mean age of 10.9 years [100].

#### 3.5.2. Environmental Factors Affecting Social Competence

Parental characteristics have also been related to social competence problems in individuals with FXS. More positive comments were associated with lower internalizing behaviors in adults, males, and females [84]. Higher closeness in the relationship with the mother was associated with lower levels of withdrawal and lower anxious/depressed behavior at a trend level in girls aged between 10 and 15 years [95]. Maternal responsivity predicted socialization scores in infants with FXS with or without autism at 30 months, with socialization scores increasing by 0.03 for every point increase in frequency of maternal responsivity behaviors, and maintaining high levels of maternal responsivity reduced the amount of decline exhibited in socialization scores [93]. Strikingly, maternal flexibility was associated with anxious/depressed behavior in their offspring, with higher flexibility associated with higher anxious/depressed behavior [78]. 

Furthermore, three studies have reported parental mental issues associated with social competence factors. Higher parental psychopathology was associated with internalizing problems in boys aged between 6 and 17 years [66], and higher maternal depressive symptoms between families were associated with higher internalizing symptoms [86]. Higher maternal psychological distress was associated with higher withdrawal levels in girls aged between 10 and 15 years [95].

Regarding home characteristics, the mother’s marital status was associated with the probability of having a mutual friend in individuals with FXS, with adolescents and adult sons or daughters of married mothers exhibiting a lower probability of having a mutual friend [86]. Higher home environment (parent responsivity, encouragement of maturity in the child, acceptance of the child, learning materials at home, effort to provide cultural, recreational, or artistic enrichment, family companionship, and quality of physical environment) was also associated with better gaze, vocal quality, and less task avoidance in boys and girls with FXS aged between 6 and 17 years [59]. Living out of the parents’ house was significantly associated with more frequent socializing but participating less frequently in religious services and hobbies [110].

## 4. Discussion

This review summarizes results for behavior problems and social competence profiles in individuals with FXS by selecting the studies that best fit behavioral phenotypes and show the best methodology criteria. Most of the studies that controlled for ASD symptomatology and achieved a good methodological quality were developed in the social competence area, while less methodological quality was found in studies addressing behavior problems. Consequently, to facilitate carrying out this review, more studies addressing behavior problems were included, even though they had lower methodological quality scores. In terms of the age ranges included, eighteen studies assessed childhood stages (up to 10 years), twenty-eight studies assessed stages from childhood to young adulthood (up to 25 years), and five studies assessed from childhood to adulthood (over 25 years). Most of the studies reviewed that addressed changes from adolescence to adulthood encompassed the social competence area. However, there were few studies in the behavior problem section, supporting the idea of the scarcity of studies on behavior problems [51]. This Discussion section is based on all the studies assessed in the screening section but not selected due to receiving a lower score, in addition to other reviews and studies that make up the empirical body of behavior problems and social competence in FXS. 

The prevalence of behavior problems observed in this review is in general agreement with previous findings for different behavior problems. This is the case for the prevalence of attention and ADHD-related problems [114], and clinical range [115], although even higher scores were found in boys (77%) in other studies [116]; for the prevalence of aggression and severity rates [117,118], and self-injurious behavior [118,119,120,121], although higher prevalence has been found in studies with functional analyses [122]; and for the prevalence of thought problems in the clinical range [116]. Low percentages in the clinical range for delinquent behavior align with studies finding scores for this behavior in the normal range [123]. Depression scores reported here are consistent with other studies [39], although higher percentages in the clinical range in younger girls have been found [124]. Furthermore, higher percentages of withdrawn behaviors in the clinical range were found in other studies with boys (38.8%) [116] and girls (37%) [115]. Moreover, social problems in the clinical range reported in this review are below the 75% found in boys [116], are similar to a sample of girls in the same age range [115] and are higher than the 29% reported in older girls [125]. The prevalence of social avoidance reported corresponds to the well-established phenotypic characteristics [126]. Externalizing, internalizing, and total problems in the clinical range show a high variance in the studies reviewed and concur with other studies [116].

Regarding trajectories of behavior problems, the decreases in aggressive behaviors during adolescent stages may resemble those seen in the TD population and concur with results that have found an improvement in aggressive behaviors after adolescence in boys with FXS, which were drug dependent [127,128]. Interestingly, decreases in aggressive behavior depended on ASD symptomatology, decreasing less over the years in individuals with ASD comorbidity than in individuals without ASD comorbidity [78]. To our knowledge, this is the first finding signaling how autism symptomatology could mediate aggressive behavior, although higher aggressive behaviors have been reported in comorbid individuals with FXS+ASD [111]. Furthermore, decreasing trajectories in behavior problems over time affected by ASD symptomatology have been found [51].

The decreases in attention problems found in this review differ from the results of previous reviews and studies that have found stability in attention problems for individuals with FXS’ entire life [75,114,129]. A cross-sectional study partially supported these results, finding a negative association between age and inattention scores in females aged between 4 and 66 years [130]. The EXPLAIN study also found lower mean decreasing ADHD scores across participants, with individuals over 18 years old exhibiting the lowest scores [131]. Different trajectories of attention problems depending on ASD symptomatology and maternal characteristics have also been found [78]. This result is supported by the strong association between ADHD symptomatology and ASD symptoms and might expand results from studies that have found developmental improvements in cognitive attention in samples of boys aged between 4 and 7 and 4 and 10 years [128,132,133].

Contrary to previous studies reporting decreases in hyperactivity behavior [75,128,130,134,135], attention/hyperactivity and over-activity were stable over time in the studies reviewed based on samples aged between 3 and 30 years and 6 and 54 years [77,88]. A closer look at the results of one of them shows lower scores for attention/hyperactivity from primary school to adolescence, although the finding may not be significant [77]. Furthermore, the study does not report results in primary school or in young adulthood. The other study found stability over eight years for FXS-only individuals and those with comorbid ASD [88]. However, this study used a different measure to assess hyperactivity.

Regarding trajectories of social competence, mixed results were found for socialization scores in the studies reviewed, such as declines in socialization during infancy which then stabilized in adolescence [91], a trajectory of restraining improvements or decreases in the middle of childhood (aged 4–8 years overall), with the differences between individuals with FXS only and those with FXS+ASD also decreasing [93,94], and steady increases from 1 to 10 years old [92]. Supporting the finding of increases during infancy, improvements in socialization scores in boys aged 9–15 years over three assessment points have also been reported, with a higher increase from 11/13 to 13/15 years than from 9/11 to 11/13 years [134]. Three trajectories based on past research on socialization in FXS have been proposed, including declines over time, positive trajectories which then decline or stabilize, and increasing trajectories over time, suggesting that they could depend on gender, age, the measures used, or the number of time assessment points [136]. Based on the results of this review, ASD symptomatology should also be considered since it has been found to influence socialization growth at 24 months [99], while autism symptomatology mediated the increase in social skills in boys aged between 3 and 14 years [34]. Autism comorbidity was controlled in all but one of the studies reviewed [91], but in a categorical way. All three of the studies reviewed included both males and females with FXS in their samples. Therefore, further investigation should be conducted to discern the socialization profile in individuals with FXS from childhood to adolescence controlled by ASD symptomatology. Furthermore, the studies reviewed do not provide information on transitions in socialization from adolescence to adulthood. 

Regarding internalizing symptoms, and in contrast to other studies which found stability, one study found higher symptoms in adolescence than in childhood for females with FXS [84]. These results agree with those found in this review regarding social anxiety, which was mainly stable in all the studies except for two, with one pointing out increasing social anxiety problems in adolescent stages in boys, and the other decreasing social avoidance over the years [90,96]. However, in one of the studies, the association with age was not significant when receptive language was accounted for [96]. Moreover, the other study included a wide age range of individuals between 2 and 46 years [90]. Furthermore, a positive association was found between social anxiety and autism symptoms [90]. Higher scores in generalized anxiety with increasing age have also been reported up to 15 years old [89]. Nonetheless, stability in anxiety scores was reported for both males and females [128]. Further exploration into anxiety and internalizing symptoms during adolescence is therefore needed to consider variables that might affect social anxiety such as ASD symptomatology.

In terms of comparisons between samples, a worse profile for behavior problems is generally reported for individuals with FXS than for TD individuals, as would be expected, while some behaviors do not differ between the groups, including somatic complaints and delinquent behavior in boys and girls [22,66]. These behavior problems are reported in a lower prevalence in this review and do not pertain to the behavioral phenotype of individuals with FXS [31]. The lack of differences in aggressive behaviors between individuals with FXS and those with TD with bad attentional abilities [101] is supported by studies in the general population reporting that inattention scores predict elevated aggressive-disruptive behaviors [137]. In addition, other studies found that individuals with FXS are less aggressive than individuals with other IDs, such as Angelman and Smith–Magenis syndromes [138]. Furthermore, some factors have been identified as increasing the risk for aggressive behaviors, such as impulsivity and hyperactivity [18,138].

Compared to individuals with DS, differences were found in attention and hyperactivity, stereotyped behaviors, repetitive behaviors, insistence on sameness, and impulsivity, which were more prevalent in FXS individuals [101,102]. These results are supported by others that also found higher rates of inattentive behavior and hyperactivity, fewer stereotypical behaviors in individuals with DS than in individuals with FXS, and greater behavior problems for individuals with FXS than for DS individuals [139,140,141].

Compared to individuals with other IDs, differences were rarely found in behavior problems, apart from those specific and characteristic of the specific syndrome, such as autistic-related behaviors such as eye avoidance and self-absorbed behaviors [79], the prevalence of stereotypy and frequency of self-injury [80], and general activity level and repetitive behaviors [79,102]. However, other studies have reported lower aggressive behaviors than in individuals without a defined etiology and individuals with other IDs [142,143]. Fewer externalizing behaviors than in individuals with Williams–Beuren have also been found [144], as well as an equal prevalence of repetitive behaviors between FXS and Prader–Willi [142]. Therefore, a general inference could not be made due to the specificity of the syndromes.

Compared to individuals with ASD, individuals with FXS showed no differences in more significant behavior problems in terms of activity level, manic/hyperactive behaviors, and distractibility [35,73,105], in agreement with studies reporting higher levels of activation in individuals with FXS than in those with ASD [145].

Regarding social competence, and as expected considering their behavioral phenotype, individuals with FXS showed lower socialization scores that developed at lower rates and exhibited higher social avoidance and withdrawal behaviors than TD individuals. Higher social avoidance, fewer social interaction gestures, and delayed communication repair have also been found [68,146,147]. Furthermore, a low prevalence of depression for individuals with FXS has been seen, resembling the prevalence in the general population [63,148]. Higher anxiety scores for boys with FXS than for the CA- or MA-matched TD group were reported in two studies [77,101], while not in others [66]. This discrepancy could be explained by differences in the comparison groups, such as whether or not they accounted for CA or MA, whether they included individuals with siblings, or differences in the age ranges used. Furthermore, other studies have found a higher prevalence of anxiety in a sample with FXS than in the general population [38].

With regard to comparisons with DS individuals, lower socialization skills, more eye gaze avoidance, and higher social anxiety have been found for FXS individuals [96,97,104,106]. These results are supported and explained by the high levels of sociability of individuals with DS and the lower difficulties in social functioning [140,149]. 

Compared to individuals with other IDs, the only findings were that individuals with FXS were more avoidant and exhibited less withdrawal and less eye contact [90,104]. Supporting these findings, more avoidant behavior in girls with FXS and requiring more time to initiate interaction than individuals with Turner syndrome have been found [125], in addition to higher eye gaze avoidance than individuals with other IDs [150].

Compared to individuals with ASD, a better profile of social competence appears, along with higher anxiety levels for individuals with FXS [73,105,110]. This profile is supported by other studies which found higher personal-social profiles for individuals with FXS than individuals with ASD [151,152,153,154]. This may be expected considering that the primary diagnosis of ASD in the Diagnostic and Statistical Manual of Mental Disorders-5 (DSM-5; American Psychiatric Association [152]) points to specific impairments in social interactions along with other behaviors [155]. Moreover, the higher levels of anxiety are consistent with the explanation for the social deficits in individuals with FXS in contrast to individuals with ASD [48]. In FXS, social deficits are driven by high levels of anxiety and hyperarousal, while in individuals with ASD, they are driven by a lack of interest [48]. A higher social preference in individuals with FXS than in individuals with ASD has also been reported [156].

Although not unitary, most of the studies reviewed found higher behavior problems for individuals with comorbid FXS+ASD than for FXS-only individuals in different age bands (early childhood, adolescence, and adulthood), except for one that did not find any differences between samples [63]. However, in this study, the assessment of comorbidities was carried out by asking parents if their offspring had ever been treated or diagnosed for a comorbid condition such as autism, not by using a validated measure. This worse profile of behavior problems mainly comprised behaviors that could be included under externalizing behaviors [157], such as hyperactivity, aggression or irritability, and repetitive behaviors, although greater attention problems have also been reported in individuals with comorbid ASD. Other studies support this worsening profile, showing generally greater problems in males with FXS+ASD than in those with FXS only [158], in addition to increasing rates of repetitive behaviors in individuals with FXS with comorbid ASD in comparison to those with FXS only [159]. Furthermore, higher rates of challenging behavior were exhibited for comorbid individuals with FXS+ASD compared to those with FXS only [160]. Differences in attention problems and hyperactivity between both samples could be supported by the causal relation between ADHD and ASD symptoms in a sample with ASD [78,132].

The few studies addressing behavior problems in individuals with FXS+ASD compared to autistic-only individuals showed discrepancies in their results. Studies showed a worse profile for comorbid FXS+ASD in adolescent/young adult samples, while for infant samples, it was the autism group that showed higher impairments in behavior problems [35,46,47]. Since ASD behaviors usually vary over time [161], a single conclusion could not be inferred. However, the results of two studies reviewed resembled each other, showing a similar worse profile of behavior problems such as total problems, aggressive, thought, and attention for the FXS+ASD group [35,46]. Further research on these comparisons is needed to further define the behavioral phenotype of FXS.

Regarding social competence, the studies reviewed achieved a higher agreement in terms of implications of FXS+ASD comorbidity, including higher impairment in socialization scores and withdrawn/lethargy behavior [26,35,46,92,93,94,105,112]. These results agree with those of some studies suggesting that social impairments could be the greatest predictors of comorbidity [162]. Other studies have found higher withdrawn behaviors in comorbid FXS+ASD individuals than in FXS-only individuals [126], while others have found no differences in mood swings/depression when comparing groups with FXS and those with FXS+ASD [27].

Comparisons between studies focusing on individuals with comorbid FXS+ASD and those with ASD only in social competence showed contradictory results, and thus an inference could not be drawn. Further investigation should be conducted in this area to delineate differences between the two syndromes, although some studies have reported that FXS+ASD individuals are more similar to ASD individuals than FXS individuals in social competence [48].

Parental characteristics such as criticism, maternal warmth, and the number of positive comments were associated with behavior problems in individuals with FXS [35,84]. Although maternal warmth was not associated with behavior problems [78], it is essential to highlight that the measures used in the studies reviewed were different. A clear interaction of maternal flexibility with behavior problems in their offspring was found, although in an unexpected way for social anxiety [78]. Besides parental characteristics such as positive comments, closeness in the relationship, mother–offspring interaction, and maternal responsivity have been associated with social competence factors [84,93,95]. Other studies have also reported characteristics of mothers such as expressed emotion to be related to behavior problems and social competence [141]. It seems that parental characteristics influence behavior and social competence problems, but there is scant research on parental characteristics, and thus further research should be conducted to elucidate the true implication.

Furthermore, parental mental health has been associated with behavior problems and social competence, including parental psychopathology, maternal depressive symptoms, and maternal psychological distress [66,86,95], although the last variable was not associated with behavior problems [100]. Although not unitary, these results, along with previous studies [163], point out that the mothers’ interactions with their offspring might be influenced by their psychological situation, which also impacts the behavioral traits of their offspring. Further efforts should be made to discern parental mental health status and the implications for their offspring with FXS.

Regarding characteristics of the home situation, and reinforcing the positive results for living outside the parental home due to spending more time socializing [110], one study conducted under the COVID-19 lockdown found increasing behavior problems reported by mothers in that period [164].

Several limitations should be considered regarding misinterpretation or overlooked information. First, only one reviewer assessed the eligibility criteria, and second, two papers were not fully retrieved even after requesting them from the authors. Third, the search string, although broad, might not have retrieved all the relevant papers for the review, and the search was only conducted in three databases. Fourth, specific information related to autism symptoms, such as the information provided by autism questionnaires, was not included in this review because it was not relevant to its objective. Lastly, three studies reviewed specified that some mothers reporting their children’s information had the full mutation condition. Although not usually considered in studies, the mother’s condition might be relevant because it may affect the way the child is raised and the reported information provided by them about their offspring.

## 5. Conclusions

The researched profile of behavior problems in the last 20 years concurs with the prevalence of previous works replicating results from past research. However, considering behavior trajectories, recent longitudinal studies may have discovered buried developmental changes thanks to higher methodological control. Regarding the prevalence of social competence problems, the studies also provided the expected results, which were similar to those from other studies. However, fewer studies reported information about prevalence due to the nature of the construct and the specificity of some measures. Trajectories for social competence have also been affected by recent studies, which have highlighted the need to continue with longitudinal research in this specific area and focus mainly on the childhood to adolescent and adulthood stages [165]. Similar findings on the behavior problems and social competence of individuals with FXS and TD individuals reflect specific areas in which FXS individuals behave the same way as TD individuals, driven by a lower prevalence of these problems in individuals with FXS, thus further profiling the phenotype characteristics of individuals with FXS. Specific comparisons with individuals with DS are also in line with previous works, signaling a better behavior problem profile and social competence for these individuals. Regarding comparisons with ASD individuals, both a higher activation and higher social competence were seen for individuals with FXS, concurring with the previous literature. 

When addressing the phenotype of FXS, ASD comorbidity is still a controversial issue [19]. However, recent studies such as those reviewed here shed light on this area, which could benefit future interventions with comorbid FXS+ASD individuals. Notably, the addition of ASD comorbidity to the FXS profile seems to implicate higher behavior problems that are mainly related to the externalizing profile, and worse social competence for the comorbid individuals. Regarding comparisons between individuals with FXS with comorbid ASD and individuals with ASD, inferences could not be drawn due to the scarcity of research. However, it seems that these comparisons could help define the comorbid profile to a greater extent than only just relying on comparisons with individuals with FXS. Last, environmental variables coming from parental characteristics have been reported to influence behavior problems and social competence in their offspring. However, only 5 of the studies among the 38 reviewed, all of which included parent-reported information on behavior problems and social competence in individuals with FXS, reported the mother’s condition. Thus, greater effort should be made in terms of the methodology, with increased control of the variables that influence the conditions, such as ASD comorbidity or symptomatology and parental characteristics, to fully understand the phenotype of individuals with FXS.

## Figures and Tables

**Figure 1 genes-13-00280-f001:**
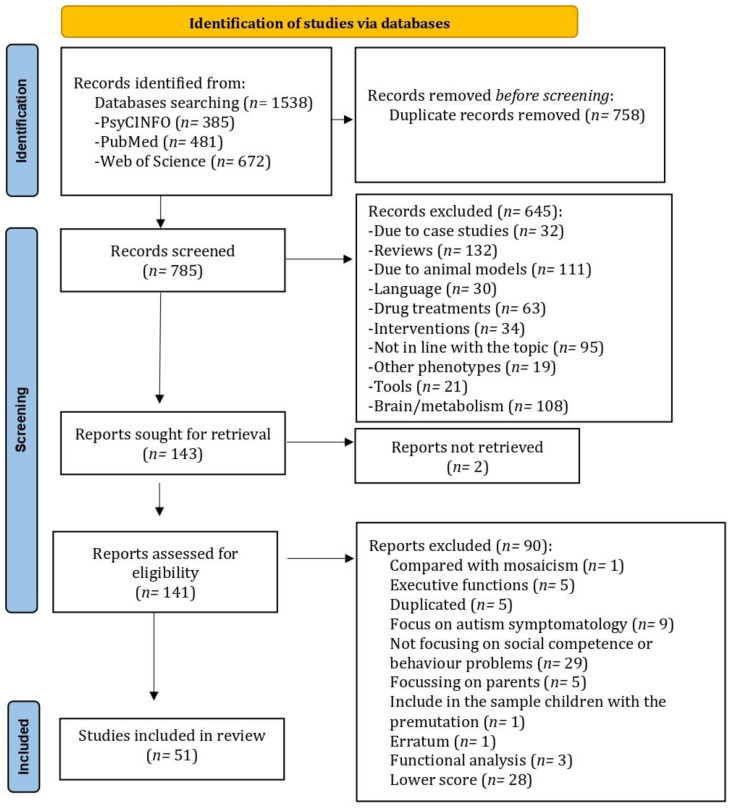
Based on PRISMA flowchart [74].

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
