# Peer review of "Behavior Problems and Social Competence in Fragile X Syndrome: A Systematic Review"

_genes, 2022, doi:10.3390/genes13020280_

Round 1

Reviewer 1 Report

The authors comprehensively studied published evidence in these 20 years in behavioral problems and social competence in FXS. The analysis included a comparison between FXS and ASD as well as developmental trajectories of the symptoms. Environmental factors such as parental characteristics which could influence behavioral and social problems have also been studied. Majority of the findings in this systematic review agree with previous evidence and reviews, while some developmental trajectories might have changed such as a developmental decrease in ADHD-like problems. This review will help researchers in the field to comprehensively understand an update of evidence in these 20 years about behavioral and social symptoms as well as developmental trajectories and environmental factors which might have a potential as effective interventions for the affected individuals in FXS. There are several minor issues needed to be revised.

1. Terminology

Unabbreviated technical terms need to be presented at the first place they appear in the text or a terminology section listing all abbreviations may be created. Given the characteristics of the journal for readers in broad fields, it is recommended. The terms include “ID” (Line 31), “CA” (Line 324), “MA” (Line 383), “IDD” (Line 401), “IQ” (Line 429), “SMS” (Line 473), “NSID” (Line 473), “Aut” (Line 531), “DLD” (Line 531), “PDD” (Line 550), “DSM-5” (Line 777), “APA” (Line 777).

Line 48 : Unabbreviated terms for ADHD should be presented here, not in Line 59.

Line 475 : Unabbreviated terms for CdLS and RTS should be presented here, not in Line 483 and 484.

There are several nonstandard technical terms used in the text. I recommend using standard ones based on commonly used databases such as HUGO or NIH.

Line 32 : “FMRP translational regulator 1” or “fragile X mental retardation 1” (HGNC, OMIM #309550, NCBI Gene ID #2332) instead of “fragile X gene” is commonly used as an unabbreviated name of “FMR1” gene.

Line 40 : “FMRP protein” or “FMRP” (HPRD ID: 02398) are recommended instead of “FMR1 protein”. In Line34, “FMRP” can be presented after “fragile X mental retardation protein” as an abbreviation.

Line 205 : “FAXTAS” appears to be a typo. “FXTAS” is an abbreviation for fragile X associated tremor/ataxia syndrome (OMIM # 300623).

2. Reference

Several statements were based solely on one paper written in Spanish. The authors may need to cite other evidence written in English which support the conclusion if any.

Line 53 – 54

Line 381 – 383

Line 389 – 390

Line 393 – 395

Line 453 – 456

3. Minor points

Line 321 : The subtitle of the section appears to be a typo. This is supposed to be “Developmental trajectories in social competence”.

Line 408 : Parenthesis seems to be missing after “difficulties with changes”.

Line 412 - 413 : “syndrome” should be added after “Phelan-McDermid”.

Line 431 : Line break appears to be missing before 3.3.3 subsection.

Line 498 : Line break appears to be missing before 3.4 subsection.

Line 527 : “than the the FXS+ASD” appears to be a typo.

Line 611 : It is better to specify what score the original article employed.

Line 702 : “T1”, “T2”, and “T3” are not necessary as these words alone do not have any specific meaning.

Line 842 – 844 : Apparently the authors’ claim is that the finding in ref [110] is reinforced by [161]. If so, it may be better to make it clearer.

Reviewer 2 Report

Cregenzan-Royo and colleagues have done an exhaustive review of all papers written about behavior problems in FXS and this is one of the most thorough reviews that I have read. There are just a few errors that need to be corrected

  1. line 64 uses the term ADH symptoms but is this ASD or ADHD?
  2.  line 70-71says among males with FXS more than 90% present typical behaviors. However what are the typical behaviors they mean?
  3.  line 155 needs a comma between anxiety and manic -hyperactive behavior.
  4. Line 207 needs the spelling of FXTAS corrected.
  5. line 206 FXS is not a premutation condition so eliminate the term FXS before premutation
  6. line215 FXS associated tremor ataxia studies is not appropriate. Use the term fragile X-associated tremor ataxia syndrome or FXTAS.
  7. line 438 sentence is incomplete but maybe this is a heading instead so have this stand out as a heading.

Overall this paper has lots of information that is somewhat exhausting to read and there is some repetition. If they can shorten some of the repetition as in social competence that would be good. If they could develop a visual about how the behavior problems change over the lifetime that would be very helpful to summarize the data. The table is very helpful too. This is an excellent paper with a very thorough review of behavior problems in FXS and it adds to the extensive literature.
